# Effect of Yakae-Prajamduen-Jamod Traditional Thai Remedy on Cognitive Impairment in an Ovariectomized Mouse Model and Its Mechanism of Action

**DOI:** 10.3390/molecules27134310

**Published:** 2022-07-05

**Authors:** Supawadee Daodee, Orawan Monthakantirat, Ariyawan Tantipongpiradet, Juthamart Maneenet, Yutthana Chotritthirong, Chantana Boonyarat, Charinya Khamphukdee, Pakakrong Kwankhao, Supaporn Pitiporn, Suresh Awale, Kinzo Matsumoto, Yaowared Chulikhit

**Affiliations:** 1Division of Pharmaceutical Chemistry, Faculty of Pharmaceutical Sciences, Khon Kaen University, Khon Kaen 40002, Thailand; csupawad@kku.ac.th (S.D.); oramon@kku.ac.th (O.M.); ariyawanps@gmail.com (A.T.); juthamart_m@kkumail.com (J.M.); yutthana_ch@kkumail.com (Y.C.); chaboo@kku.ac.th (C.B.); 2Department of Pharmacy, University of Naples Federico II, 80131 Naples, Italy; 3Division of Natural Drug Discovery, Institute of Natural Medicine, University of Toyama, 2630 Sugitani, Toyama 930-0194, Japan; suresh@inm.u-toyama.ac.jp; 4Division of Pharmacognosy and Toxicology, Faculty of Pharmaceutical Sciences, Khon Kaen University, Khon Kaen 40002, Thailand; charkh@kku.ac.th; 5Department of Pharmacy, Chao Phya Abhaibhubejhr Hospital, Ministry of Public Health, Prachinburi 25000, Thailand; pakakrong2@gmail.com (P.K.); spitiporn@yahoo.com (S.P.); 6Graduate School of Pharmaceutical Sciences, Daiichi University of Pharmacy, Fukuoka 815-8511, Japan; k-matsumoto@daiichi-cps.ac.jp

**Keywords:** cognitive impairment, ovariectomy, Yakae-Prajamduen-Jamod, neuroinflammation, oxidative stress, HPA axis

## Abstract

Cognitive impairment is a neurological symptom caused by reduced estrogen levels in menopausal women. The Thai traditional medicine, Yakae-Prajamduen-Jamod (YPJ), is a formula consisting of 23 medicinal herbs and has long been used to treat menopausal symptoms in Thailand. In the present study, we investigated the effects of YPJ on cognitive deficits and its underlying mechanisms of action in ovariectomized (OVX) mice, an animal model of menopause. OVX mice showed cognitive deficits in the Y-maze, the novel object recognition test, and the Morris water maze. The serum corticosterone (CORT) level was significantly increased in OVX mice. Superoxide dismutase and catalase activities were reduced, while the mRNA expression of IL-1β, IL-6, and TNF-α inflammatory cytokines were up-regulated in the frontal cortex and hippocampus of OVX mice. These alterations were attenuated by daily treatment with either YPJ or 17β-estradiol. HPLC analysis revealed that YPJ contained antioxidant and phytoestrogen constituents including gallic acid, myricetin, quercetin, luteolin, genistein, and coumestrol. These results suggest that YPJ exerts its ameliorative effects on OVX-induced cognitive deficits in part by mitigating HPA axis overactivation, neuroinflammation, and oxidative brain damage. Therefore, YPJ may be a novel alternative therapeutic medicine suitable for the treatment of cognitive deficits during the menopausal transition.

## 1. Introduction

Menopause is a normal physiological event described by the final menstrual period. It is caused by the declension of ovarian follicular activity and results in reduced estrogen [1]. Estrogen plays an important role in maintaining brain homeostasis and function. This ovarian hormone promotes mitochondrial bioenergetics, protects against oxidative damage, and decreases the formation of amyloid beta (Aβ) peptides by upregulating the expression of Aβ-degrading enzymes [2,3,4]. Estrogen also induces spinogenesis and synaptogenesis upon the enhancement of fast glutamatergic transmission and long-term potentiation (LTP) [4]. Estrogen exhibits neuroprotective properties that are mediated via the increased expression and activity of proteins involved in oxidative phosphorylation, including ATP synthase and pyruvate dehydrogenase [2]. Estrogen deprivation has been shown to disrupt the balance between free radical production and the antioxidant defensive system in the brain, leading to oxidative stress and neuronal cell injury. The consequential brain damage results in many undesirable symptoms, such as cognitive impairment, sleep disorders, anxiety, and depression. These symptoms are frequent in women who have had their ovaries removed or have reached menopausal age [5,6,7]. On the other hand, several studies have proven that estrogen and testosterone regulate HPA axis responses by buffering glucocorticoid feedback [8]. Evidence indicates that estrogen depletion induces the elevation of serum corticosterone, which confirms the prevalence of heightened stress. Chronic corticosterone exposure impaired memory retention in ovariectomized female rats [9]. In addition, a large body of evidence also demonstrates that estrogen exerts anti-inflammatory activity, which may be linked to the inhibition of microglia activation. The loss of ovarian estrogens in OVX rodents may exacerbate central and peripheral inflammatory responses by promoting the release of a large number of neuroinflammatory cytokine levels, such as IL-1, IL-6, TNF-α, and activating immune system cells involved in the regulation of neurogenesis, synaptic plasticity, and survival of neuronal cells. These changes, in turn, are associated with cognitive dysfunctions [10,11]. Epidemiological analyses indicate that providing estrogen therapy during the menopause transition can reduce the risk of cognitive decline in postmenopausal women [12], and that women not receiving estrogen replacement therapy following oophorectomy-induced menopause are at increased risk for cognitive impairment [13]. However, estrogen therapy has been shown to increase the risk of breast cancer and endometrial cancer [13,14] and is contraindicated for women with a known or suspected history of these cancers. Therefore, there is considerable interest in developing herbal medicines or herbal remedies as alternatives to estrogen therapy [5].

Traditional Thai medicine (TTM) has been utilized for the diagnosis, prevention, and treatment of illness for a long time. TTM is a holistic medicinal approach based on indigenous theories, beliefs, and experiences. It integrates knowledge from traditional Chinese medicines, Ayurveda, folk Thai culture, spiritual healing, and astrology [15,16]. In TTM, various single herb and traditional Thai herbal formulae, such as *Pueraria candollei* [17], *Asparagus racemosus Willd* [18], and Yakae-Prajamduen-Jamod (YJP) remedy, are used as alternatives for the treatment of menopausal symptoms [19]. YPJ is a well-known TTM used for the treatment of peri-menopausal and menopausal symptoms in Thailand. YPJ consists of 23 medicinal herbs, as shown in Table 1. YPJ capsules have been prescribed by Thai folk doctors at the Chao Phya Abhaibhubejhr hospital (the central public hospital in Prachinburi) for more than 50 years [19]. Our previous study demonstrated that the YPJ formula attenuated ovariectomy-induced anxiety and lipid peroxidation in the hippocampus and frontal cortex of ICR mice [20]. However, the YPJ remedy has not been investigated in a mouse model of estrogen deprivation associated with cognitive impairment related to brain oxidative damage and neuro-inflammation. Therefore, the present study aimed to investigate the effects of YPJ administration on cognitive behavior in ovariectomized mice as a model of estrogen deprivation. The effects of YPJ on serum CORT level, antioxidant enzyme activity, and inflammatory cytokine-related gene expression (IL-1β, IL-6, and TNF-α) were examined to determine the putative molecular mechanisms of YPJ formula activity.

## 2. Results

### 2.1. Effect of YPJ Remedy on OVX-Induced Cognitive Deficit-like Behavior

Estrogen deprivation has been shown to induce cognitive impairment. The Y-maze test, the novel object recognition test (NORT), and the Morris water maze test (MWMT) were used in this study to determine whether YPJ powder ameliorates OVX-induced learning and memory impairment. 

The Y-maze test is a recognition test for assessing spatial memory function that is based on the normal navigational behavior of mice. The proportion of Y-maze arm entries made by mice without repetition is calculated as the percentage of spontaneous alternation, and higher scores indicate that the mice remember their previous choices [21]. As shown in Figure 1, vehicle-treated OVX mice exhibited a significantly lower percentage of spontaneous alternation than sham mice, indicating an OVX-induced spatial memory deficit. Treatment with either YPJ powder at a dose of 500 mg/kg/day or 17β-estradiol (E_2_, 1 μg/kg/day) significantly increased the percentage of spontaneous alternation when compared to vehicle-treated OVX mice (for detailed statistical analysis, see Appendix A).

The NORT is a tool for assessing memory function based on instinctive murine behavior to prefer a novel object to a familiar object. Object recognition is represented by the percentage discrimination index, which measures the difference in time spent exploring novel and familiar objects [22]. Higher percentage discrimination index values indicate that more time was spent exploring the novel object. The vehicle-treated OVX mice had a lower percentage of discrimination index values compared to the sham group, indicating that they failed to recognize and remember objects (for detailed statistical analysis, see Appendix A). Daily treatment of OVX mice with either estradiol (1 μg/kg/day) or the YPJ formula (100 and 500 mg/kg/day) significantly increased the percentage of the discrimination index compared to vehicle-treated OVX mice, indicating improved memory function (Figure 2).

The Morris water maze test (MWMT) is a widely used test for assessing spatial learning and memory in animal models [23]. In the visible trial conducted on the first day, there were no significant differences between groups in the time mice spent finding the platform, indicating no motivational or sensory–motor deficits. There was a significant decrease in escape latency time observed in the sham group during the training phase, showing that the sham mice learned the location of the submerged platform. In contrast, vehicle-treated OVX mice showed no reduction in escape latency time, indicating significantly impaired spatial learning and memory performance (for detailed statistical analysis, see Appendix A). Daily treatment with either YPJ or 17β-estradiol significantly reduced escape latency for OVX mice on days 3, 4, and 5 (for detailed statistical analysis, see Appendix A). The results of the probe test session conducted on day 6 (Figure 3B) showed that vehicle-treated OVX mice spent significantly less time in the target quadrant than did the sham mice, indicating that ovariectomy caused learning and memory deficit. Oral administration of either 500 mg/kg/day YPJ or 1 μg/kg/day 17β-estradiol to OVX mice significantly increased the time they spent in the target quadrant, indicating improved learning and memory performance (for detailed statistical analysis, see Appendix A).

### 2.2. Effect of YPJ Remedy on Serum CORT Level

Serum CORT levels were examined in order to clarify whether the negative feedback in the regulation of the hypothalamic–pituitary–adrenal axis (HPA axis) was impaired by estrogen deprivation. Figure 4 shows that the vehicle-treated OVX mice had significantly higher serum CORT levels than the sham-operated mice. This OVX-induced elevation of serum CORT levels was significantly suppressed by daily treatment with either YPJ or 17β-estradiol (for detailed statistical analysis, see Appendix A).

### 2.3. Effect of YPJ Remedy on Antioxidant Enzyme Activities in the Frontal Cortex and Hippocampus 

The effects of YPJ formula on catalase (CAT) and superoxide dismutase (SOD) antioxidant enzyme activities in the brains of OVX mice were determined. The vehicle-treated OVX mice showed significantly reduced CAT and SOD activities in both the frontal cortex and hippocampus compared to the sham group. The administration of estradiol (1 μg/kg/day) significantly increased CAT and SOD activities in both the frontal cortex and hippocampus. In the same way, mice treated with 500 mg/kg/day YPJ also showed significantly elevated CAT and SOD activities in both the frontal cortex and hippocampus compared with vehicle-treated OVX mice (Figure 5) (for detailed statistical analysis, see Appendix A).

### 2.4. Effect of YPJ Remedy on OVX-Induced Changes in Inflammatory Cytokine mRNA Expression in the Frontal Cortex and Hippocampus

Quantitative real-time PCR analysis showed that vehicle-treated OVX mice had significantly elevated interleukin 1β (IL-1β), interleukin 6 (IL-6), and tumor necrosis factor α (TNF-α) mRNA expression in the frontal cortex and hippocampus compared with the sham group. Estradiol (1 μg/kg/day) and YPJ (500 mg/kg/day) both significantly reduced the OVX-induced increases in IL-1β, IL-6, and TNF-α mRNA expression (Figure 6) (for detailed statistical analysis, see Appendix A).

### 2.5. High Performance Liquid Chromatography (HPLC) Analysis of Constituents of YPJ Extract and Validation of the Method

The major constituents of YPJ formula are flavonoids, phenolic acids, polyphenolic compounds, isoflavones, and coumestans. Gallic acid (**1**), myricetin (**2**), quercetin (**3**), luteolin (**4**), genistein (**5**), and coumestrol (**6**) were used as markers for HPLC analysis of the YPJ extract (Figure 7). The HPLC method was validated for the analysis of these six compounds at a concentration range of 1–10 μg/mL. The validation parameters determined were range, linearity, limit of detection (LOD), limit of quantitation (LOQ), precision, and accuracy.

The validation results are presented in Appendix A. The HPLC method showed good accuracy and linearity, with a coefficient of determination (R^2^) greater than 0.99 across the concentration range of 1–10 μg/m. The limit of detection (LOD) and limit of quantitation (LOQ) validation data corresponded to signal-to-noise ratios equal to 3 and 10, respectively. The percentage recoveries were between 90 and 110%. The inter-day (within-day) and intra-day (between-day) precision values were both less than 2% of the percentage relative standard deviation (%RSD). Representative chromatograms of the standards and YPJ are shown in the Appendix A, respectively. The amounts of each of the six compounds present in the ethanolic extract of YPJ were: genistein (0.503 ± 0.057 mg/g extract), coumestrol (0.536 ± 0.031 mg/g extract), quercetin (0.784 ± 0.039 mg/g extract), myricetin (1.467 ± 0.038 mg/g extract), gallic acid (1.799 ± 0.120 mg/g extract), and luteolin (3.605 ± 0.479 mg/g extract).

## 3. Discussion

The traditional Thai remedy YPJ consists of 23 herbal plants (Table 1). YPJ is well-known for the treatment of peri-menopausal and menopausal symptoms in Thailand. Our previous study demonstrated that the YPJ formula ameliorated anxiety-like behavior in OVX mice and had high antioxidant activity [20]. Taken together, these ethno-medical applications and the anti-oxidative profile of YPJ suggest that this traditional remedy has good potential for the treatment of cognitive dysfunction due to estrogen deprivation. Hence, this current investigation used an OVX mouse model to evaluate the anti-dementia-like effects of the YPJ formula and elucidate its mechanisms of action. In the current study, OVX induced cognitive impairment in mice and increased oxidative stress, serum CORT level, and expression of the inflammatory-related genes IL-1β, IL-6, and TNF-α in the frontal cortex and hippocampus. Daily administration of the YPJ remedy ameliorated cognitive dysfunction by reversing the OVX-induced reduction of antioxidant enzyme activities, HPA axis dysregulation, and neuro-inflammation in the hippocampus and frontal cortex. HPLC fingerprint analysis identified gallic acid, myricetin, quercetin, luteolin, genistein, and coumestrol in the YPJ formula. The memory-enhancing effects of the YPJ remedy could be due to the pharmacological activities of these constituents.

Over the last three decades, it has become apparent that estrogen exerts effects on brain function that go well beyond the regulation of reproductive functions. In fact, this ovarian hormone facilitates several higher-order cognitive performances by exerting effects on the hippocampus and prefrontal cortex [8,24]. Estrogen also plays an important role in synaptogenesis and spinogenesis by initiating a complex set of signal transduction pathways via membrane-bounded estrogen receptors that are highly expressed in excitatory synapses in the hippocampus. Rapid estrogen signaling leads to axospinous synapse formation in the hippocampus through the combined stimulation of filopodia formation in the neuron and activation of Akt (protein kinase B) via PI3K signal transduction. Activated Akt then initiates translation of postsynaptic density protein 95 (PSD-95), a scaffold protein of postsynaptic densities in excitatory neurons, to produce synapse size enhancement [24]. There is also evidence that E2-dependent upregulation of brain-derived neurotrophic factor (BDNF) and the subsequent activation of tropomyosin kinase receptor B (TrkB) promotes hippocampal synaptic plasticity, which enhances learning and memory [25,26]. On the other hand, estrogen has been shown to regulate intracellular brain metabolism, glucose transport, oxidative phosphorylation, and ATP formation in mitochondria. In parallel, estrogen also increases antioxidant defenses by up-regulating several antioxidant enzymes, including peroxiredoxin 5, glutaredoxin, and manganese superoxide dismutase (MnSOD), thereby reducing free radicals and lowering oxidative damage to brain mitochondria [22,25]. Therefore, the estrogen loss induced by menopause or ovariectomy can exacerbate the effects of aging on cognitive decline [27,28,29]. From a basic science perspective, ovariectomy is the gold standard procedure for understanding estrogen deficiency through animal experiments.

In the present study, OVX-induced cognitive decline was evaluated by three different tests, including the Y-maze test, NORT, and MWMT. The Y-maze test and MWMT evaluate spatial memory, whereas the NORT evaluates recognition/working memory. The Y-maze test evaluates short-term memory by allowing mice to explore all three arms of the Y-maze apparatus and is based on the natural desire of mice to explore new areas. A rodent with good frontal cortex and hippocampus functions will recall the arms it has recently visited, indicating good recall memory [30]. The MWMT assesses hippocampal-dependent spatial navigation memory, a form of declarative memory that involves the ability to learn to utilize and remember distal landmarks [31]. The NORT assesses the spontaneous tendency of rodents to spend more time exploring a novel object than a familiar one [32]. In the current study, estrogen deprivation for 6 weeks following OVX produced cognitive dysfunction, as shown by a decrease in spontaneous alternations in the Y-maze test, a failure to discriminate between novel and familiar objects in the NORT, and an increase in escape latency time and a decrease in time spent in the target quadrant in the MWMT. These learning and memory impairments are in agreement with those noted in previous finding [6,29,33].

Interestingly, our data demonstrate that daily treatment with either 17β-estradiol or YPJ significantly improved spatial and non-spatial memories in an animal model of menopause/ovariectomy. In order to illuminate the molecular mechanism(s) by which the administration of YPJ remedy ameliorated cognitive impairment in OVX mice, we investigated antioxidant enzyme activity, serum CORT level, and inflammatory gene expression in the hippocampus and frontal cortex. Spatial and non-spatial working memories reliant on the dorsolateral PFC and hippocampus are susceptible to decline with aging and menopause, and this may be partly due to the high energy requirement to sustain this function [25]. Mitochondria have a central role in energy production for neurons, where energy requirements are especially high. Several lines of evidence indicate that estrogen is a fundamental regulator of mitochondrial energy production [27]. Thus, ovariectomy induces a significant reduction in estrogen, which in turn leads to the impairment of mitochondrial energy metabolism and oxidative phosphorylation. Mitochondrial dysfunction is often closely associated with the overproduction of free radicals and reduced antioxidant enzyme activities [34,35]. The resulting oxidative stress has been widely associated with the development and progression of neuronal cell damage, which in turn leads to cognitive dysfunction in estrogen-deficient postmenopausal women and Alzheimer’s disease [29,34,35,36]. In the present study, ovariectomy led to a reduction in the activities of superoxide dismutase (SOD) and catalase (CAT) in the frontal cortex and hippocampus of mice in a manner that was prevented by treatment with either 17β-estradiol or YPJ. Therefore, it is plausible that the anti-dementia effects of YPJ in OVX mice are mediated by the chemical constituents in YPJ remedy that prevent these OVX-induced reductions in SOD and CAT activities in the frontal cortex and hippocampus, thereby preventing neuronal damage and ameliorating cognitive impairment.

Gonadal steroids profoundly influence brain development and function and are also apparently responsible for the regulation of the HPA system. Estrogen potentiates and buffers glucocorticoid feedback via the regulation of corticotropin-releasing hormone (CRH) expression in the hypothalamus and corticosteroid receptor Type I (MR) and Type II (GR) expression in the hippocampus [8,37]. The elevated serum CORT levels observed in the current study confirmed that the OVX mice were experiencing heightened stress. Elevated serum CORT levels are characteristic of aged human subjects, indicating that the HPA axis is activated. Central disturbances in the HPA negative feedback regulatory system are believed to be responsible for the age-associated increase in CORT titers. This HPA over-activation is related to decreased sensitivity of glucocorticoid receptors and failure to suppress the secretion of adrenocorticotropic hormone (ACTH), thereby contributing to the dysregulation of CORT secretion from the adrenals [9,37]. The hippocampus contains a high amount of GC receptors and is highly sensitive to the negative effects of GCs. Chronic GC exposure results in dendritic retraction, neuropil and neuronal cell damage, and impairments in cognitive performance [37,38,39]. In addition to exacerbating damage by other insults, continual GC exposure and estrogen deprivation have been shown to upregulate pro-inflammatory cytokines, such as interleukin-1β (IL-1β) and interleukin-6 (IL-6); activate microglia; and damage various brain regions [27,40,41]. Increases in microglial markers in response to this type of extreme GC exposure and estrogen deficiency indicate increased neuro-inflammation as part of the immune response [37]. Consistent with previous findings, increased levels of immune activation were shown in this study through augmented levels of TNF-α, IL-1β, and IL-6 mRNA expression in the hippocampus and frontal cortex of OVX mice [41,42,43].

Reversal of the effects of estrogen deprivation in OVX mice with either 17β-estradiol or YPJ also had a mitigating effect on CORT levels, although they were not restored to the ovary-intact level. The underlying mechanism may involve improvement of the negative feedback loop on the HPA axis. Estrogenic upregulation of GC receptors allows the brain to detect better lower levels of GC, thereby making the HPA axis more responsive and resulting in more rapid breaking of the stress response [36]. This, in turn, spares the hippocampus and other brain regions, such as the prefrontal cortex, from further damage. The data from the current study show that YPJ produced significant reductions in TNF-α, IL-1β, and IL-6 mRNA expression in both brain regions of the OVX mice.

All 23 plants in the YPJ formula possess estrogenic and/or antioxidant activity. Therefore, gallic acid, a phenolic antioxidant compound, and quercetin, luteolin, myricetin, genistein, and coumestrol phytoestrogens were selected as markers for the phytochemical analysis of YPJ [5,44,45,46]. Phytochemical investigation of the YPJ ethanolic extract by HPLC revealed the presence of flavonoids and phenolic acids as the major constituents. The phytochemical content in the extract was found to be in the following order: luteolin > gallic acid > myricetin > quercetin > coumestrol > genistein. Luteolin is an active constituent found in *Carthamus tinctorius* L., *Phyllanthus emblica, Senna siamea, Coriandrum sativum*, *Cyperus rotundus, Nigella sativa*, *Saussurea lappa*, and *Artemisia annua* [47,48,49,50,51,52,53,54,55]. Luteolin has been shown to have preventive and therapeutic value for neurodegenerative diseases, including cognitive decline in elderly patients, which has been linked to its phytoestrogen-like activities, antioxidant activity, and ability to relieve neuroinflammation via suppression of microglial activation in the brain [56,57,58]. Myricetin, which belongs to the phytoestrogen and antioxidant group of flavonols, is found in many medical herbs, such as *Dracaena loureiri*. *Phyllanthus emblica*, *Nigella sativa*, and *Saussurea lappa* [48,53,54,59]. It has been reported that myricetin has favorable effects on cognitive performance. The underlying mechanisms are believed to be mediated by myricetin’s influence on BDNF signaling in the hippocampus, action of inhibiting acetylcholinesterase (AChE), and its antioxidant properties [60]. Quercetin has several promising bioactive effects such as anti-inflammatory, antioxidant, anti-Alzheimer’s, and estrogenic-like activity [61,62]. Quercetin supplementation was able to ameliorate cognitive impairment in OVX mice by restoring histone acetlytransferase (HAT)/histone deacetylase (HDAC) homeostasis through ERK stimulation and reversing modifications in neuroplasticity markers in the cortex and hippocampus [63]. Genistein exerts an effect on hippocampal synaptogenesis via several complementary mechanisms involving ERα and ERβ as well as G protein-coupled estrogen receptor (GPR30) [5,64,65]. GPR30 has a relatively high binding affinity for genistein. The activation of GPR30 produces the enhancement of CA1 spine density and hippocampal synaptic transmission [66]. In addition, genistein and quercetin exhibit positive allosteric modulatory (PAM) effects at the α7 nicotinic acetylcholine receptor (nAChR) [61]. This receptor is highly expressed in the hippocampus, cortex, and several subcortical limbic regions. nAChR also plays important roles in cognition, concentration, and working memory [61,64,65,66]. Thus, allosteric potentiation of α7 may be an additional mechanism underlying the neuroprotective actions of flavonoids. Coumestrol is a phytoestrogen in the class of phytochemicals known as coumestans found in *Derris* spp. [67]. Coumestrol binds to estrogen receptors (ERα and ERβ) with relatively high affinity [68], demonstrating that coumestrol improved mitochondrial function in an OVX Wistar rat model due to its modulatory effects on mitochondrial respiration and reduction of brain oxidative stress [69].

Considering the threat associated with estrogen replacement therapy (ERT) on menopausal women, it is controversial since estrogen increases the risk of breast cancer, endometrial cancer, and venous thromboembolic events. Consequently, the interest in medicinal herbs containing phytoestrogen has increased due to the realization that ERT is neither as safe nor as effective as previously speculated [70]. The findings in this study suggest that the YPJ remedy is a TTM that contains estrogenic compounds. The efficacy of YPJ in overcoming estrogen deprivation-induced oxidative stress, HPA axis hyperactivation, and neuroinflammation was shown. Apparently, the YPJ remedy is a safe alternative to ERT in alleviating learning and memory deficits in menopausal women.

## 4. Materials and Methods

### 4.1. Plant Materials and YPJ Preparations

The YPJ remedy was provided by Chao Phya Abhaibhubejhr Hospital, Prachinburi Province, Thailand. This remedy consists of 23 medicinal plants, as shown in Table 1. Plant materials were identified by Benjawan Leenin, chief of the Traditional Knowledge Center, Chao Phya Abhaibhubejhr Hospital Foundation. The vouchered specimens were deposited at the museum of Chao Phya Abhaibhubejhr Hospital.

All plant ingredients were cleaned, sliced into small pieces, dried at 50 °C in a hot air oven, and powdered. The ingredients were weighed and mixed according to the proportion shown in Table 1. The YPJ remedy powder was macerated with 95% EtOH at room temperature for 72 h, filtered through Whatman No.1 filter paper, and re-macerated twice. Extracts were combined and dried using a rotary evaporator under reduced pressure at 50 °C. The YPJ extract was kept at −20 °C throughout the experiment.

### 4.2. Animals

Sixty female ICR mice (four weeks old, weighing 20–30 g) were obtained from the National Laboratory Animal Center (Mahidol University, Nakhon Pathom, Thailand). All mice were housed on wood chip bedding in cages with food and water ad libitum. Housing conditions were 12-h dark and light cycle (light 06.00 a.m.–06.00 p.m.) under temperature control (22 ± 2 °C) and constant humidity (45 ± 2%) in the Laboratory Animal Unit of the Faculty of Pharmaceutical Sciences, Khon Kaen University. The experiment protocols in this study were conducted in accordance with the Guiding Principles for the Care and Use of Animals (NIH Publications No. 80–23, revised in 1996) and were approved by the Animal Ethics Committee of Khon Kaen University (ACUC-KKU-54/2559, Reference No. 0514.1.75/60).

### 4.3. Surgical Procedure and Drug Administration

Surgical operation of OVX in mice was conducted as previously described [20,71]. Briefly, the mice were anesthetized by injecting sodium pentobarbital (Nembutal^®^: Ceva Santa Animale, Libourne, France) at a dose of 60 mg/kg (i.p.). OVX animals underwent bilateral ovariectomy by a dorsolateral incision. The subcutaneous connective tissue was closed using sterile chromic catgut sutures (VIGIKENZ^®^, Penang, Malaysia). The operation wound was cleaned with 70% ethanol and povidone–iodine solution (BETADINE^®^, Stamford, CT, USA). The mice were randomly separated into twelve mice in each group. The sham-operated animals underwent all surgical operations without removing the ovaries. After a 3-day recovery period, OVX animals were divided into four groups: (1) ovariectomy (OVX), (2) ovariectomy + 1 μg/kg 17β-estradiol (OVX+E2), (3) ovariectomy + 100 mg/kg YPJ (OVX + YPJ100), and (4) ovariectomy + 500 mg/kg YPJ (OVX + YPJ500). The sham and OVX groups were orally administered corn oil (vehicle). 17β-Estradiol was intraperitoneally administered. YPJ powder was suspended in corn oil and orally administered once daily for 8 weeks. On behavioral testing days, administration was conducted 1 h before the behavioral assessment. The YPJ dose was derived from the clinical dose (1600 mg/day) prescribed in the hospital. This dose was converted into the appropriate dose for mice by the following equation: human equivalent dose (HED, mg/kg) = mouse dose (mg/kg) × mouse Km/Human Km), where Km is the correction factor [20]. The experiment framework is summarized in Figure 8.

### 4.4. Behavioral Studies

#### 4.4.1. Y-Maze Test

The Y-maze apparatus consisted of three black polyethylene arms of equal size (3 cm × 40 cm × 18 cm) oriented at 60° angles. The Y-maze test was conducted 1 h after drug treatment. The mice were placed on one arm, and the sequence of arm entries was observed and recorded manually over an 8 min testing period [29,39]. An alternation was defined as an entry into all three arms without a repeat entry. Percentage alternation was calculated using the following equation:%Alternation = [(Number of alternations)/(Total arm entries − 2)] × 100

The entire apparatus was cleaned with 70% ethanol to remove odor cues between sessions.

#### 4.4.2. Novel Object Recognition Test (NORT)

The novel object recognition test (NORT) uses rodents’ natural proclivity for exploring novelty to test recognition memory. The NORT was performed as described previously [29,39]. Briefly, 24 h before the test, mice were habituated to an empty examination box (50 × 50 × 40 cm high) for 10 min. The NORT then consisted of two sessions, a sample phase trial and test phase trial that were separated by a 30 min interval. During the sample phase trial, mice were placed in the examination box with 2 identical objects (A2 and A2) in the corner at a specified distance from each other (15 cm from each adjacent wall). The mice were then free to explore the area for 5 min. The amount of time that the animal spent exploring each object was measured. Exploring time was defined by animal nose-point detection within a 2 cm radius around the object. During the test phase trial, one of the previously explored objects was replaced with a novel object, and the time spent exploring each object was recorded during a 5-min observation period. Recognition memory was assessed by measuring the animal’s ability to recognize an object previously displayed. The percentage of discrimination index was calculated according to the following equation:%Discrimination Index = [(TN − TF)/(TN + TF)] × 100
where TN = time devoted to the novel object, TF = time devoted to the familiar object. Higher % discrimination index values represent better recognition memory. The experimental apparatus, including the objects, were cleaned with 70% ethanol to prevent a build-up of olfactory cues between the sessions.

#### 4.4.3. Morris Water Maze Test (MWMT)

The Morris water maze test is used to evaluate the spatial reference memory of animals and was conducted as previously described [29,39]. The Morris water maze was composed of a black circular pool (68-cm-diameter and 18-cm-depth) filled with water (temperature around 25 ± 1 °C). A dark platform (4 × 12 × 14 cm) was placed 1 cm below the water surface in the middle of target quadrant (Q1) to render it invisible to the mouse. Mice received four trials daily over a 5-day period in the training phase. In each trial, the mouse was introduced into various quadrants of the pool (Q1–Q4) facing the wall of the tank and allowed to swim for 60 s in search of the hidden platform. The time spent to reach the hidden platform was recorded as “escape latency”. Mice that failed to find the position of the hidden platform within 60 s were manually guided to the platform by an experimenter. The mouse was permitted to stand on the hidden stage for 10 s before being brought to a plastic chamber for 60 s before the next trial. Data for each day were averaged over the 4 trials for result analysis. For the probe test, the hidden platform was removed from the pool, and a single 60 s trial was performed to assess how well the mice had learned the location of the platform. The time that the mouse spent swimming in the target quadrant (Q1) was recorded as the index of reference memory.

### 4.5. Determination of Serum Corticosterone Level

At the conclusion of the behavioral testing, mice were anesthetized with pentobarbital sodium (Nembutal^®^: 60 mg/kg, i.p.; Ceva Sante Animale, Libourne, France), and blood samples were collected via cardiac puncture immediately after decapitation. Blood samples were left at room temperature and centrifuged at 3000 rpm at 21 °C for 20 min. The CORT level in serum was measured using a commercial CORT ELISA test kit (Assaypro LLC., St. Charies, MO, USA). The serum CORT levels were calculated according to the previously described procedure [70].

### 4.6. Determination of Antioxidant Enzyme Activities

Superoxide dismutase (SOD) catalyzes the dismutation of the superoxide anion (O_2_^•−^) into hydrogen peroxide and molecular oxygen. Catalase (CAT) catalyzes the decomposition of hydrogen peroxide (H_2_O_2_) to water and oxygen. The accumulation of O_2_^•−^ and H_2_O_2_ causes the oxidation of cellular targets leading to oxidative damage [27,29]. Removal of O_2_^•−^ and H_2_O_2_ by SOD and CAT, respectively, can protect cells from oxidative damage. The antioxidant enzyme activities of SOD and CAT in the hippocampus and frontal cortex were evaluated in this study. The hippocampus and frontal cortex were homogenized in cold phosphate buffer (5 mM, pH 7.4) to obtain 20% brain homogenates. SOD and CAT activities were measured using commercial assay kits (19160 SOD Assay Kit and CAT100 Catalase Assay Kit, Sigma-Aldrich, St. Louis, MO, USA). All antioxidant enzyme activities were normalized to the total protein concentration in the sample as determined by the Bradford method [39].

### 4.7. Quantitative Real-Time Polymerase Chain Reaction (Q-PCR)

Total RNA from the frontal cortex and hippocampus was extracted by using TRIzol^®^ reagent (Invitrogen, Rockville, MD, USA) following the manufacturer’s protocol. First-strand cDNA was synthesized with oligo(dT) primers and SuperScript III Reverse Transcriptase (Invitrogen, Rockville, MD, USA). QPCR was conducted used SsoAdvanced^TM^ Universal SYBR^®^ Green Supermix (Biorad, Hercules, CA, USA). The following primers were synthesized by Macrogen (Seoul, South Korea): GAPDH, 5′-AAC ACA GTC CAT GCC ATC AC-3′ (sense) and 5′-TCC ACC ACC CTG TTG CTG TA-3′ (antisense); IL-1β, 5′-GAC AGC AA GTG ATA GGC C-3′(sense) and 5′-CGT CGG CAA TGT ATG TGT TGG-3′ (antisense); IL-6, 5′-CTT CCA TCC AGT TGC CTT CTT-3′ (sense) and 5′-AAT TAA GCC TCC GAC TTC TGA AG-3′ (antisense); and TNF-α, 5′-GCC TCT TCT CAT TCC TGC TTG-3′ (sense) and 5′-CTG ATG AGA GGG AGG CCA TT-3′(antisense). GAPDH mRNA was used as a control to normalize expression. The data were presented as fold difference relative expression [29].

### 4.8. HPLC Analysis and Validation

#### 4.8.1. Standards and Reagents

Gallic acid, myricetin, quercetin, luteolin, genistein, and coumestrol HPLC-grade standards were purchased from Sigma-Aldrich (St. Louis, MO, USA). The HPLC-grade solvents, methanol and acetic acid, were purchased from Merck (Darmstadt, Germany). Hydrochloric acid (HCl) was obtained from RCI Labscan (Bangkok, Thailand). Tertiary-butylhydroquinone (TBHQ) and 3-tert-butyl-4-hydroxyanisloe (BHA) were purchased from Fluka (Munich, Germany).

#### 4.8.2. Hydrolysis of the Yakae-Prajamduen-Jamod Extract

The YPJ extract was hydrolyzed as previously described [72]. Briefly, 50 mg of YPJ extract was mixed with 5 mg TBHQ and 5 mg BHA. The mixtures were refluxed with 1.2 M HCl in 50% methanol at 80 °C for 2 h. Mixtures were sonicated for 30 min and filtered with 0.45 μm syringe filters before injection into the HPLC apparatus.

#### 4.8.3. HPLC Analysis and Validation

A reversed-phase HPLC system with a UV detector (Agilent Technologies Inc., Santa Clara, CA, USA) and a Hypersil ODS column (4 × 250 mm, 5 μm) was used for analyzing the YPJ extract and the gallic acid, myricetin, quercetin, luteolin, genistein, and coumestrol standards. Ultrapure water (solvent A) and 80% methanol containing 0.25% acetic acid (solvent B) was used as the mobile phase by gradient elution for 50 min at a flow rate of 2.5 mL/min and with detection at 254, 280, 340, and 370 nm.

Calibration curves were prepared from 1, 2, 4, 6.8, and 10 μg/mL dilutions of 1 mg/mL stock solutions of the gallic acid, myricetin, quercetin, luteolin, genistein, and coumestrol standards diluted with methanol. The analytical HPLC methods were validated following ICH guidelines. The linearity was validated by using linear regression analysis to calculate the coefficient of determination (R^2^) of the standard calibration curve (1–10 μg/mL, *n* = 5). The limit of detection (LOD) and limit of quantitation (LOQ) were also determined, and the signal level of the substance reached at least 3 and 10 times the signal-to-noise ratios (*n* = 10). The accuracy was measured by the percentage recovery of three standard concentrations in 5 replicates. Precision (%RSD) was validated for within-day and between-day determinations (*n* = 5).

### 4.9. Statistical Analysis

Data were expressed as the mean ± S.E.M. and analyzed by one-way analysis of variance (ANOVA) followed by the Tukey test for multiple comparisons among different groups. Differences with *p* < 0.05 were considered significant. The software SigmaStat^®^ ver 3.5 (SYSTAT Software Inc., Richmond, CA, USA) was used for data analysis.

## 5. Conclusions

YPJ remedy enhanced cognitive performance in a mouse model of dementia with estrogen deficiency by mitigating neuroinflammation and brain oxidative damage in the hippocampus and prefrontal cortex and consequently the negative feedback system of the HPA axis.

## Figures and Tables

**Figure 1 molecules-27-04310-f001:**
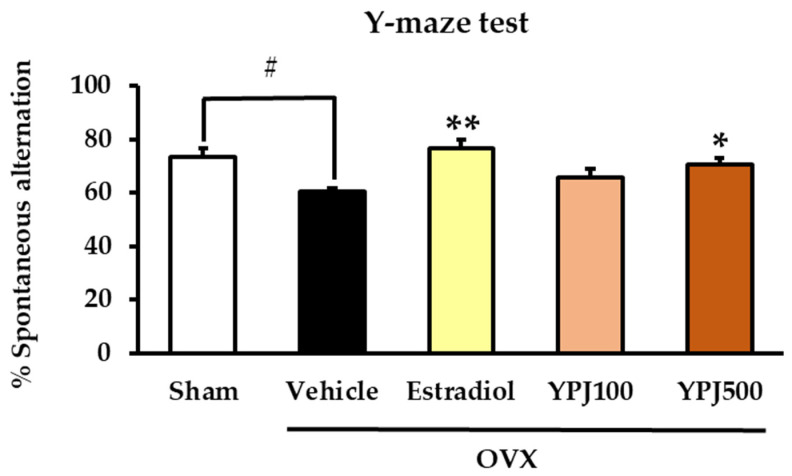
The effect of YPJ formula on cognitive performance in sham and OVX mice using the Y-maze test. Each column represents the mean ± SEM (*n* = 10). # *p* < 0.001 vs. the sham group. * *p* < 0.05, ** *p* < 0.001 vs. the vehicle-treated OVX group (post hoc Tukey test).

**Figure 2 molecules-27-04310-f002:**
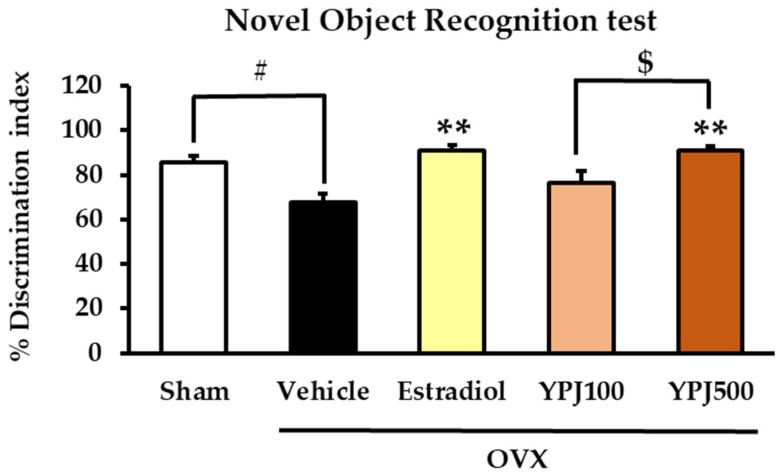
The effect of the YPJ formula on object recognition memory in the NORT. Percentage of discrimination index values in each experimental group are expressed as the mean ± SEM (*n* = 10). # *p* < 0.05 vs. the sham group. ** *p* < 0.01 vs. the vehicle-treated OVX group. ^$^ *p* < 0.05 between YPJ treatment (post hoc Tukey test).

**Figure 3 molecules-27-04310-f003:**
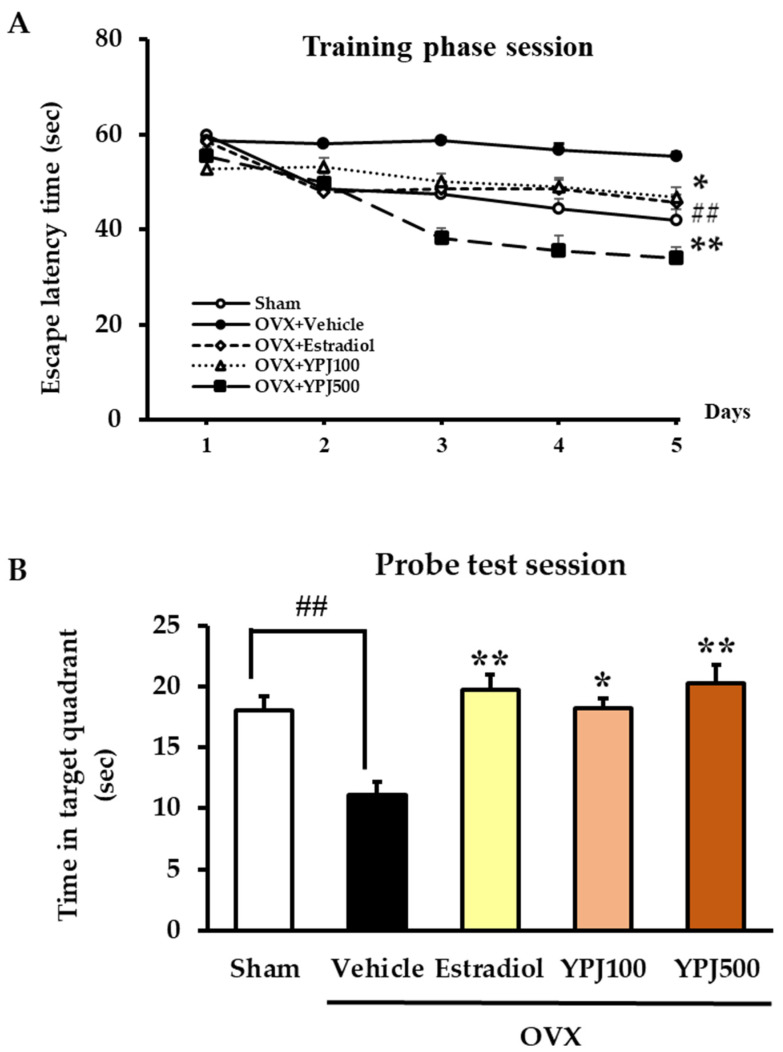
Effect of YPJ and 17β-estradiol on Morris water maze performance of OVX mice. (**A**) Learning achievement of the animals was analyzed during the training phase sessions. Data represent the mean ± SEM of escape latency times, 4 trials/day for 5 days. (**B**) Memory retrieval performance was examined in the probe test session. Data represent the mean time spent in the target quadrant ± S.E.M. (*n* = 10). ## *p* < 0.01 vs. the vehicle-treated sham group. * *p* < 0.05, ** *p* < 0.01 vs. the vehicle-treated OVX mice (post hoc Tukey test).

**Figure 4 molecules-27-04310-f004:**
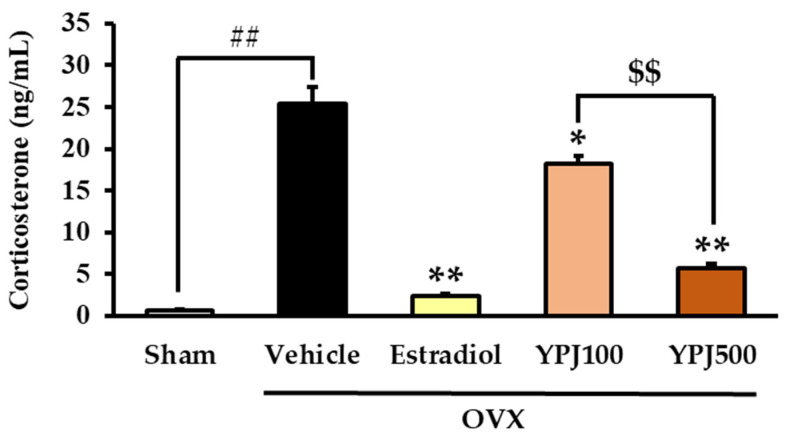
Effect of YPJ and 17β-estradiol treatment on serum CORT levels in OVX mice. Each column represents the mean ± S.E.M. (*n* = 5). ## *p* < 0.01 vs. the vehicle-treated sham group. * *p* < 0.05, ** *p* < 0.01 vs. the vehicle-treated OVX mice. ^$$^ *p* < 0.01 compared between doses of YPJ formula (post hoc Tukey test).

**Figure 5 molecules-27-04310-f005:**
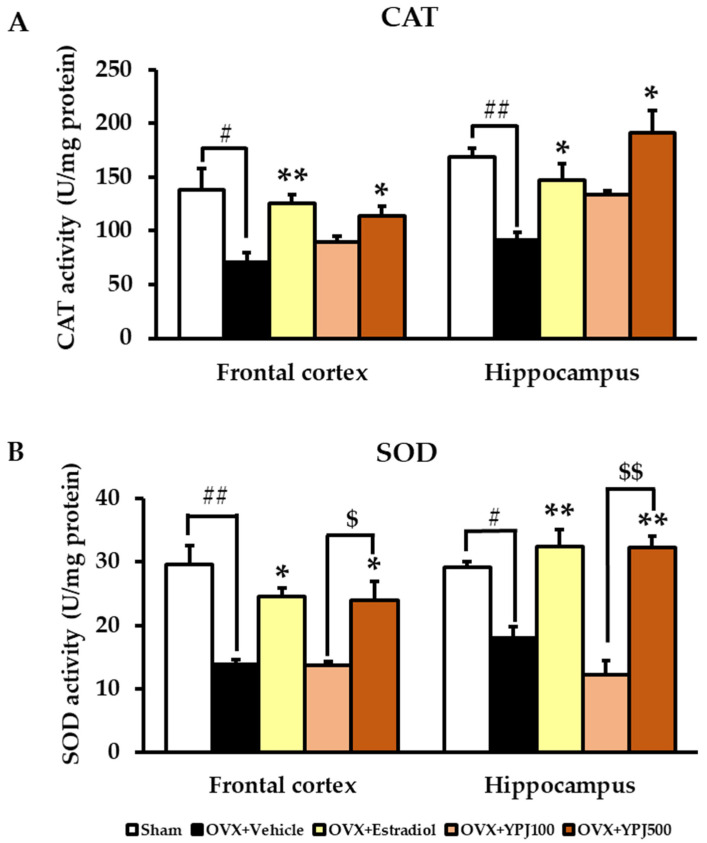
The effect of YPJ formula on antioxidant enzyme activities in the brains of OVX mice. (**A**) catalase (CAT) activity, (**B**) superoxide dismutase (SOD) activity. Data represent the mean ± SEM (*n* = 5). Significant effects are represented by # *p* < 0.05, ## *p* < 0.001 vs. sham, * *p* < 0.05, ** *p* < 0.001 vs. OVX vehicle-treated mice, ^$^ *p* < 0.05, ^$$^ *p* < 0.001 between doses of YPJ (post hoc Tukey test).

**Figure 6 molecules-27-04310-f006:**
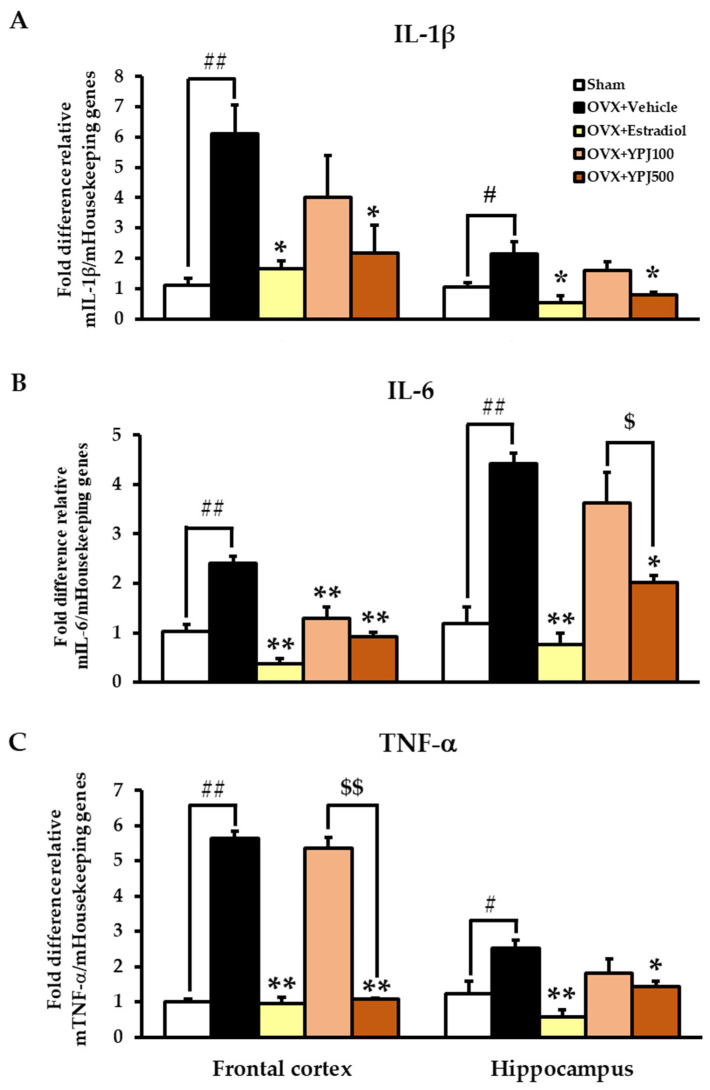
The effect of YPJ formula on inflammatory cytokine expression in the frontal cortex and hippocampus of OVX mice. (**A**) IL-1β, (**B**) IL-6, and (**C**) TNF-α. Data represent the mean ± SEM (*n* = 5). Significant effects are represented # *p* < 0.05, ## *p* < 0.001 vs. sham, * *p* < 0.05, ** *p* < 0.001 vs. vehicle-treated OVX mice, ^$^ *p* < 0.05, ^$$^ *p* < 0.001 between doses of YPJ (post hoc Tukey test).

**Figure 7 molecules-27-04310-f007:**
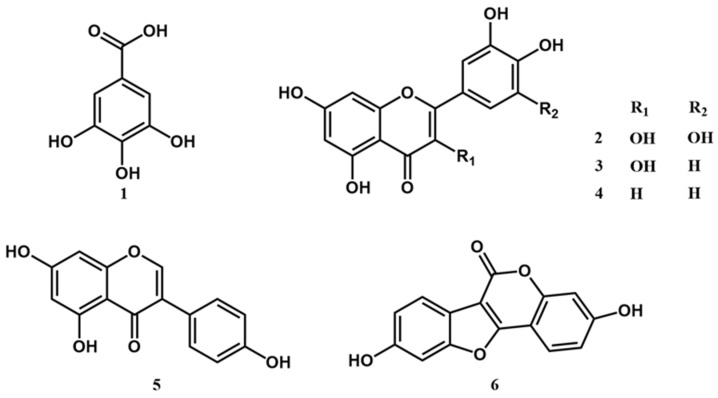
Structures of gallic acid (**1**), myricetin (**2**), quercetin (**3**), luteolin (**4**), genistein (**5**), and coumestrol (**6**).

**Figure 8 molecules-27-04310-f008:**
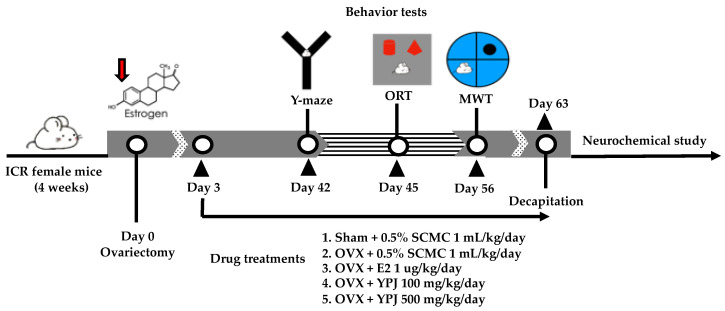
A schematic drawing of the experimental protocol.

**Table 1 molecules-27-04310-t001:** List of medicinal plants in YPJ remedy, part used, and proportion.

Plant Species (Family)	Family Name	Voucher Specimen	Part Used	Proportion
*Dracaena loureiri* Gagnep	Asparagaceae	YPJ001	heartwood	5.56
*Tarenna hoaensis* Pitard	Rubiaceae	YPJ002	heartwood	5.56
*Bridelia ovata* Decne.	Phyllanthaceae	YPJ003	leaves	5.56
*Carthamus tinctorius* L.	Asteraceae	YPJ004	flower	5.56
*Terminalia chebula* Retz. var. *chebula.*	Combretaceae	YPJ005	fruits	5.56
*Terminalia arjuna* Roxb.	Combretaceae	YPJ006	fruits	5.56
*Phyllanthus emblica* L.	Phyllanthaceae	YPJ007	fruits	5.56
*Aloe vera* (L.) Burm.f.	Asphodelaceae	YPJ008	resin	5.56
*Senna garrettiana* (Craib) H.S.Irwin and Barneby	Fabaceae	YPJ009	heartwood	3.70
*Senna siamea* (Lam.) H.S.Irwin and Barneby	Leguminosae	YPJ010	heartwood	3.70
*Derris scandens* (Roxb.) Benth	Fabaceae	YPJ011	stem	3.70
*Caesalpinia sappan* L.	Leguminosae	YPJ012	heartwood	3.70
*Mesua ferrea* L.	Calophyllaceae	YPJ013	flower	3.70
*Mammea siamensis* Kosterm	Calophyllaceae	YPJ014	flower	3.70
*Coriandrum sativum* L.	Apiaceae	YPJ015	fruits	3.70
*Myristica fragrans* Houtt.	Myristicaceae	YPJ016	fruits	3.70
*Amomum testaceum* Ridl.	Zingiberaceae	YPJ017	fruits	3.70
*Cyperus rotundus* Linn	Cyperaceae	YPJ018	tuber	3.70
*Nigella sativa* L.	Ranunculaceae	YPJ019	seed	3.70
*Piper ribesoides* Wall.	Piperaceae	YPJ020	stem	3.70
*Saussurea lappa* Clarke	Asteraceae	YPJ021	root	3.70
*Artemisia annua* L.	Asteraceae	YPJ022	whole	3.70
*Angelica sinensis*	Apiaceae	YPJ023	root	3.70

## Data Availability

Data sharing is not applicable to this article.

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
