# Peer review of "Effect of Yakae-Prajamduen-Jamod Traditional Thai Remedy on Cognitive Impairment in an Ovariectomized Mouse Model and Its Mechanism of Action"

_molecules, 2022, doi:10.3390/molecules27134310_

Round 1

Reviewer 1 Report

This manuscript studies the effect of a traditional Thai medicine used for menopausal women, YPJ, on cognitive deficits in ovariectomized (OVX) mice. Two doses of YPJ, 100mg/kg/day and 500mg/kg/day, were used in the study and estradiol treatment was used as positive control. As evaluated by Y-maze, novel object recognition and Morris water maze, YPJ significantly improved both spatial and non-spatial memories in OVX mice to a similar extent as estradiol. To understand the potential underlying mechanism, serum cortisone level, antioxidant enzyme activity and inflammation markers were measured. YPJ significantly reduced serum cortisone level, elevated catalase and SOD enzymatic activity in the brain and repressed the gene expression of some inflammation markers including IL-6, IL1b and TNFalpha in the frontal cortex and hippocampus. 

The manuscript is well-written and the data presentation is clear. 

Suggested minor changes:

1. In the introduction, authors could include background of other herbal medicines (in addition to YPJ) as a alternate to estrogen therapy.

2. Line 141, HPA axis is not defined.

Author Response

Thank you very much for your valuable comments on our manuscript. We clearly understand the reviewer’s comments. These comments are very useful to improve our manuscript. We have revised the manuscript according to your suggestions.

Suggested minor changes:

  1. In the introduction, authors could include background of other herbal medicines (in addition to YPJ) as an alternate to estrogen therapy.

Response: Thank you for your suggestion. The background of some herbal medicines as the alternatives to estrogen therapy have been added in the introduction section.

  1. Line 141, HPA axis is not defined.

Response: Hypothalamic-pituitary-adrenal axis (HPA axis) is already defined.

Reviewer 2 Report

The authors investigated the effects of Yakae-Prajamduen-Jamod Traditional Thai Remedy on Cognitive impairment:

My comments are as follows:

1.      In need of a citation in the following statement to proof that YPJ has been indeed used as folk medicines/remedies.

YPJ capsules have been prescribed by Thai folk doctors at Chao Phya Abhaibhubejhr hospital (the central public hospital in Prachinburi) for more than 50 years.

2.      To restructure the main aim as:

However, the YPJ remedy has not been investigated in a mice model of estrogen deprivation associated with cognitive impairment related to brain oxidative damage and neuro-inflammation

3.      No significant difference was observed between 500mg/kg YPJ and 17b-estradiol in SOD and CAT activities; IL-1β, IL-6 and TNF-a expressions.

How would the authors support the therapeutic benefits of 500mg/kg YPJ over estradiol?

4.      Line 38 – neuroinflammation

5.      How many mice per group?

6.  100 mg/kg and 500 mg/kg of YPJ were administered orally. Did the authors conduct a pilot study to obtain a dose-response curve for toxicity evaluation? This is important considering YPJ is a multiblend remedy being fed to the mice for a period of ~57 days.

7.      The study did not test the effects of bioactive compounds. The statement shall be removed from the Introduction

8.      To provide the link between estrogen deprivation and neuroinflammation, and between estrogen deprivation and dementia.

9.    Only three biomarkers were tested to justify its anti-neuroinflammatory potential.

(i) Microglial activation and neuroinflammatory pathways should be validated as well

(ii) In addition, immunohistochemistry analysis of brain samples shall be carried out as well.

10. Lack of focus on the investigation of different aspects of cognitive impairments. Are the following aspects related to estrogen deprivation? - neuroinflammation, oxidative stress or dementia?  The bioassays do not seem to be sufficient to address each of these aspects.

Author Response

Response to Reviewer 2

Reviewer #2

The authors investigated the effects of Yakae-Prajamduen-Jamod Traditional Thai Remedy on Cognitive impairment:

My comments are as follows:

  1. In need of a citation in the following statement to proof that YPJ has been indeed used as folk medicines/remedies.

YPJ capsules have been prescribed by Thai folk doctors at Chao Phya Abhaibhubejhr hospital (the central public hospital in Prachinburi) for more than 50 years.

Response: We have added the citation.

  1. To restructure the main aim as:

However, the YPJ remedy has not been investigated in a mice model of estrogen deprivation associated with cognitive impairment related to brain oxidative damage and neuro-inflammation

Response: We have rephrased the sentence.

  1. No significant difference was observed between 500mg/kg YPJ and 17b-estradiol in SOD and CAT activities; IL-1β, IL-6 and TNF-aexpressions.

How would the authors support the therapeutic benefits of 500mg/kg YPJ over estradiol?

Response: We have added the content in the last paragraph of discussion part.

  1. Line 38 – neuroinflammation

Response: Correction is made.

  1. How many mice per group?

Response: Twelve mice per group.

                 We have added this information in “4.3. Surgical Procedure and Drug    Administration”.

  1. 100 mg/kg and 500 mg/kg of YPJ were administered orally. Did the authors conduct a pilot study to obtain a dose-response curve for toxicity evaluation? This is important considering YPJ is a multiblend remedy being fed to the mice for a period of ~57 days.

Response: We wish to thank reviewer #2 for his/her comment. We did not perform a pilot study to obtain a dose-response curve for toxicity evaluation. We did only the acute oral toxicity test. The acute oral toxicity (Class method) of YPJ was assessed in Sprague-Dawley male rats according to OECD Guideline 423 with a single-dose exposure to determine a clearly toxic dose and to assign the LD50 dose. A sufficient number of dose levels viz. 5 mg/kg, 50 mg/kg, 300 mg/kg, 2,000 mg/kg and 5,000 mg/kg were applied to determine the approximate lethal dose. Clinical observations after feeding the rat for 24 hours and 14 days revealed no mortality or abnormalities. According to our study, YPJ can be classified into category 5 (LD50 cut-off value 5,000 mg/kg), according to the Globally Harmonized System for the classification of chemicals.The dose of YPJ remedy in this study was calculated from the human equivalent dose (HED) equation.

In addition, in clinical practice, Thai folk doctors in Chao Phya Abhaibhubejhr hospital usually prescribe YPJ 1-2 capsules, twice daily, for 2-6 months in order to treat the menopausal symptoms.

  1. The study did not test the effects of bioactive compounds. The statement shall be removed from the Introduction

Response: We have rephrased the sentence.

  1. To provide the link between estrogen deprivation and neuroinflammation, and between estrogen deprivation and dementia.

Response: We have added the link between estrogen deprivation and neuroinflammation and function of HPA axis. In addition we also have added the link between estrogen deprivation and dementia in the introduction section.

  1. Only three biomarkers were tested to justify its anti-neuroinflammatory potential.

(i) Microglial activation and neuroinflammatory pathways should be validated as well

(ii) In addition, immunohistochemistry analysis of brain samples shall be carried out as well.

Response: We wish to thank reviewer #2 for calling our attention for the immunohistochemistry analysis of brain samples and the pathway of microglia activation.

Estradiol has specific actions on microglia and astrocytes. After a proinflammatory stimulus, nuclear factor kappa-light-chain-enhancer of activated B cells (NF-κB) is translocated to the nucleus to activate genes of different proinflammatory pathways. This spreads the damage in the brain by increasing proinflammatory mediators. In microglia, estradiol inhibits the release of metalloproteinase 9 (MMP9). In addition, by activating PI3K, NF-κB translocation to the nucleus is inhibited. This inhibition will lead to a decrease in the production of nitric oxide synthases (iNOS), which will subsequently reduce the production of nitric oxide (NO) and reactive oxygen species (ROS). Both NO and ROS are responsible for expanding the inflammatory response when damage occurs, so its inhibition has an anti-inflammatory effect [1]. Estradiol will also bind to ERβ and GPER1/GPR30 microglial receptors, thus modulating the release of inflammatory mediators and reducing microglial activation [2,3].

In addition to the anti-inflammatory aspect of estradiol on microglia, it has also been described that the hormone is capable of modulating the phagocytic capacity of microglia during development [2,3] and in animal models of Alzheimer’s disease. Thus, it has been widely observed that estradiol exerts protective actions in animal models of Alzheimer’s disease.

Regarding to the reviewer suggestions, there are many biomarkers of neuroinflammation associate with estrogen deficiency. Therefore, the further study will be performed in order to clarify the detail of anti-neuroinflammatory potential of YPJ remedy.

References

  1. Baker, A.E.; Brautigam, V.M.; Watters, J.J. Estrogen modulates microglial inflammatory mediator production via interactions with estrogen receptor beta. Endocrinology2004145, 5021–5032. [Google Scholar] [CrossRef] [PubMed]
  2. Vegeto, E.; Benedusi, V.; Maggi, A. Estrogen anti-inflammatory activity in brain: A therapeutic opportunity for menopause and neurodegenerative diseases.  Neuroendocrinol.200829, 507–519. [Google Scholar] [CrossRef] [PubMed]
  3. Crespo-Castrillo A.; Arevalo M.A. Microglial and Astrocytic Function in Physiological and Pathological Conditions: Estrogenic Modulation. Int J Mol Sci. 2020 May 2;21(9):3219

  1. Lack of focus on the investigation of different aspects of cognitive impairments. Are the following aspects related to estrogen deprivation? - neuroinflammation, oxidative stress or dementia?  The bioassays do not seem to be sufficient to address each of these aspects.

Response: In this study, we focus on the investigation of different aspects of estrogen deprivation-induced cognitive impairments in OVX mouse model including oxidative stress, neuroinflammation and hyperactivation of HPA axis. We conducted the Y-maze test, NORT and MWMT in order to confirm the cognitive impairment in OVX mice. We performed the determination of SOD and CAT activities in frontal cortex and hippocampus in order to investigate the effect of YPJ remedy on brain oxidative stress. Quantitative real time PCR was conducted to investigate the mRNA expression of IL-1β, IL-6 and TNF-α in frontal cortex and hippocampus. The serum CORT level was also determined in order to investigate the HPA function aspect. Therefore, the bioassays in the present study are sufficient to address the molecular mechanisms of YPJ remedy in the different aspects.

Thanks again for your valuable comments on our manuscript. These comments are very useful to improve our manuscript. We have revised the manuscript according to your suggestions.

Round 2

Reviewer 2 Report

None